# The Impacts of the Hierarchical Medical System on National Health Insurance on the Resident’s Health Seeking Behavior in Taiwan: A Case Study on the Policy to Reduce Hospital Visits

**DOI:** 10.3390/ijerph16173167

**Published:** 2019-08-30

**Authors:** Yu-Hua Yan, Chih-Ming Kung, Horng-Ming Yeh

**Affiliations:** 1Department of Medical Research, Tainan Municipal Hospital (Managed by Show Chwan Medical Care Corporation), No. 670, Chung Te Road, Tainan City 701, Taiwan; 2Department of Hospital and Health Care Administration, Chang Jung Christian University, No.60, Sec. 1, Erren Rd., Rende District, Tainan City 71710, Taiwan; 3Department of Health Care Administration, Chia Nan University of Pharmacy and Science, No.1, Changda Rd., Gueiren District, Tainan City 71101, Taiwan; 4Department of Information Technology and Communication, Shih Chien University Kaohsiung Campus, No. 200 University Road Neimen, Kaohsiung 84550, Taiwan; 5Department of Internal Medicine, Tainan Municipal Hospital (Managed by Show Chwan Medical Care Corporation), No. 670, Chung Te Road, Tainan City 701, Taiwan

**Keywords:** hierarchical medical system, national health insurance, healthcare-seeking behavior, reduction in hospital visits

## Abstract

*Objective*: This study investigated the impacts of the hierarchical medical system under the national health insurance program on residents’ healthcare-seeking behavior in Taiwan. *Background*: Healthcare authorities in Taiwan initiated an allowance reduction for outpatient visits at regional hospitals and higher hierarchical hospitals in 2018. The ultimate goal is to implement a hierarchical medical system to provide residents accessible as well as consistent medical services. *Methods*: This research was conducted through a questionnaire survey, and data were collected between August and December 2018 from the records of subjects who had recently sought medical attention. A total of 1340 valid questionnaires were returned. *Results*: A principal finding was that there were significant differences in the knowledge of new policies by age, marital status, annual income, education level, and occupation (*p* < 0.001). Regarding the effects on healthcare-seeking behavior, there were significant differences from persons aged 40–49 years (*p* < 0.1), in junior high school (*p* < 0.05), not aware of the policy (*p* < 0.001), and awareness of both the hierarchical medical system and the policy to reduce outpatient visits to large hospitals (*p* < 0.001). *Conclusion*: The health administration authorities should devote more effort into promoting knowledge of the policy in order to better inform the public about the hierarchical medical system.

## 1. Introduction

Taiwan’s current healthcare system is divided into four levels: Medical centers, regional hospitals, district hospitals, and clinics. After the implementation of the National Health Insurance (NHI) program in 1995, the convenience of seeking different healthcare facilities was based on the accessibility of healthcare services. The current NHI program discourages unnecessary healthcare-seeking behavior due to greater patient choice, easy access, and low medical cost burden [1]. Unnecessary healthcare-seeking behavior also leads to a waste of medical resources [2]. The personal care expenditure of the NHI increased from NT 252 billion in 1998 to 595.3 billion in 2016, indicating a growth of 2.36 times. Healthcare expenditure by clinics has also increased, from 162.7 billion points in 1998 to 392.4 points in 2016, a growth of 141.2%. The average healthcare cost per person per visit has also risen annually. From the trend of NHI revenues and expenditures, safety reserves are expected to be exhausted by 2021, and Taiwan will face pressure to adjust NHI premiums, a challenge that is exacerbated because the country is now in an economic recession. The income levels of employees and the insured are not increasing, and there are limits to the growth of the gross domestic product. If prices are constantly rising, the public and hospitals must jointly reduce expenditures on unnecessary healthcare in order to avoid increases in NHI premiums [3].

When the NHI program was introduced, the essence of the “medical referral system and hierarchical medical system” was included in the National Health Insurance Act. After the implementation of the NHI program, hospital crowding regardless of disease severity resulted in packed hospitals and overcrowded emergency rooms; meanwhile, district hospitals were shutting down [4,5]. According to the number of outpatient visits at each healthcare level, in 2006, 70.2% of personal healthcare expenditures were used in clinics of basic Western medicine, while 11.2, 10.2, and 8.4% went to district hospitals, regional hospitals, and medical centers, respectively. In 2016, however, personal healthcare expenditures in clinics of Western medicine and district hospitals decreased by 64.7 and 9.7%, respectively, while outpatient visits to regional hospitals and medical centers increased by 14.8 and 10.8%, respectively [3].

Hence, the government of Taiwan has promoted a hierarchical medical system to enable medical centers and regional hospitals to refocus on the fundamental areas of teaching, research, and care for emergency cases and difficult diseases. Since 2017, the growth in the numbers of outpatient visits for minor ailment services at medical centers and regional hospitals has been limited; the number of outpatient visits could not exceed 90% of those in 2016 or the NHI program would not reimburse the costs. As a result, the upcoding of diagnosis for minor ailments by hospitals to reduce the number of patients under primary care was instituted to respond to the cuts in reimbursement. As these limits were not effective, in 2018, the Ministry of Health and Welfare established a hierarchical medical system and medical referral systems to promote reduced outpatient visits to large-scale hospitals to cut outpatient visits to medical centers and regional hospitals by 2% in 2018 and to reach the target of a 10% reduction within five years. Those facilities that fail to reach the target will not be reimbursed by the NHI program [6]. Therefore, policy marketing is critical to policy implementation [7].

The hierarchical medical system implemented internationally requires restricting patients’ freedom of choice [8]. The optimal allocation of medical and healthcare resources improves public health, accessibility to care, and healthcare efficiency, and may even resolve health inequalities [9,10]. With regard to the current NHI policy in Taiwan, however, reform is highly disrupted by political forces, and no one is willing to engage in reform [11]. Taiwanese citizens can see any doctor without a referral. They may also go to any level of hospital directly as they wish [5]. The patients have unlimited rights of free choice of physicians and healthcare facilities [12]. Therefore, patients are encouraged to engage in potential doctor shopping and receive only fragmented healthcare services [13]. To enable each level’s healthcare institution to return to a normal medical ecosystem and to entice patients with minor ailments to visit local clinics, we have to gradually implement a hierarchical medical system and bi-directional referral systems [14].

The hierarchical medical system has become an essential system in many developed countries [8]. Currently many countries, such as Germany, allow upgraded healthcare access and face greater healthcare expenditures than other countries, such as the United Kingdom, that require referral to a clinic [15,16]. Brown and Theoharides [17] found that the differential health insurance policies influenced the choice of hospital, and in particular, that patients increasingly chose higher-level hospitals. Those with high consuming power are more likely to choose to visit medical centers, and they may also go directly to any level of hospital as they wish. However, larger, more popular hospitals can become overcrowded [5]. Similarly, patients with lower consumption power would prefer to visit local clinics, a situation that does not fit with the relative equality principle of social welfare and is against the policy goal of social insurance. In addition, when patients with minor ailments are accustomed to visiting large hospitals, they are less likely to trust small hospitals or clinics when there is an emergency, which reflects a distrust in local clinics [18]. A hierarchical medical system was set up to solve the problems of biased resource allocation and high patient flows to large hospitals [8].

The NHI program in Taiwan guarantees the right for people to seek healthcare regardless of their financial status and the freedom to choose healthcare providers [19,20,21]. Because of fierce competition and fees for services in the system, medical institutions aim to increase the number of cases attended to and look for any means to achieve an advantage, such as purchasing high-tech and expensive equipment [22]. Furthermore, healthcare-seeking behavior generally refers to the freedom of patients to choose their own hospitals or physicians [23]. Individual healthcare-seeking patterns show complex interrelationships between socio-economic factors and the physical environment along with individual characteristics and behaviors [24]. While healthcare-seeking behavior traditionally was conceptualized as a “sequence of remedial actions” taken to rectify “perceived ill-health”, nowadays a wider perspective of affirmative, health-promoting behaviors has been adopted [25]. Factors affecting healthcare-seeking behavior include socio-demographic factors, social structures, level of education, cultural beliefs and practices, economic and political systems, environmental conditions, disease patterns, and health care systems [26]. Berkowitz and Flexner [27] suggested four factors that influence healthcare-seeking behavior—care quality, cleanliness of facilities, service attitude, and hospital reputation—while Boscarino and Steiber [28] categorized the factors of consideration in the order of convenience, physician, professionalism, and facilities.

Healthcare authorities in Taiwan initiated the allowance reduction of outpatient visits at regional and higher hierarchical hospitals in 2018, according to which the allowance of outpatient visits at regional hospitals and medical centers will be reduced by 2%, followed by a total reduction of 10% over five years [6]. The ultimate goal is to implement a hierarchical medical system and provide the residents accessible as well as consistent medical services. Research investigating the novel policy of “allowance reduction for outpatient visits” in Taiwan is still lacking. Most previous studies focused on the referral system rather than the effects of the hierarchical medical system on healthcare-seeking behavior [21,29]. This study investigated the effect of the hierarchical medical system under the NHI program on healthcare-seeking behavior in Taiwan, and especially focused on the Policy to Reduce Hospital Visits. The study has two objectives: (1) Understanding Taiwanese people’s awareness of the hierarchical medical system and policy to reduce hospital visits, and (2) studying the association between the novel policy and choice of medical care. The results were analyzed with a multiple regression model to further understand the actual performance of the NHI policy in order that they might serve as a reference for the government when promoting the hierarchical medical system in the future.

## 2. Materials and Methods

### 2.1. Research Sample and Source of Data

The research adopted a cross-sectional study in which patient data were collected from a sample of hospitals. The research participants comprised patients from five similar-sized regional hospitals in Taiwan. These hospitals provided similar clinical services and had similar numbers of patients. The study involved two steps. The first sought to establish content validity of the newly derived instrument. Expert validation was conducted to evaluate the content validity of the research questionnaire, and modifications were made according to expert opinion; then internal consistency was tested to check questionnaire reliability. The 30 research participants were collected from these five hospitals. The survey was administered between 1 July 2018 and 31 July 2018, and Cronbach’s α fell in the acceptable range around 0.8.

In the main study, patients were asked for consent verbally before the questionnaires were distributed. All participants were outpatients of the five hospitals at the time of the survey, and convenience sampling was used. Data were collected via face-to-face interviews. During the questionnaire distribution, if any subject was unwilling to participate in or was not qualified for the research, interviewers looked for the next eligible subject following the sampling procedure. To train the interviewers, three students from the Department of Hospital Administration were hired, and a half-day training session was conducted before the questionnaire survey. A total of 1400 questionnaires were given out. A total of 1340 valid responses were collected for a 95.7% (1340 of 1400) valid response rate, and the survey was administered and returned between August and December 2018. The study design was reviewed and approved by the Institutional Review Board (IRB No. 1070403).

### 2.2. Questionnaire Framework and Design

This study examined the impacts of the hierarchical medical system on healthcare-seeking behavior in Taiwan. This study adopted the arguments of Berkowitz and Flexner [27] and used a self-designed questionnaire whose items were scored on a 5-point Likert scale from 1 = very much disagree to 5 = very much agree. In line with our study concept, we divided healthcare-seeking behavior into the following categories: Convenience, physician, professionalism, facilities, and policy to reduce hospital visits. Thus, the questionnaire possessed specific validity with respect to the concepts and measurement of variables. Of the 32 original questions in the utility scale, we revised 6 (Appendix A). Personal attributes included the variables of gender, age, marital status, educational background, and occupation. The structured questionnaire was used to conduct face-to-face interviews with research subjects. Cronbach’s α was used to assess the reliability of healthcare-seeking behavior; Cronbach’s α = 0.826, indicating high reliability.

### 2.3. Data Processing and Analysis Method

This study used SPSS 18.0 to conduct data analysis, and in addition to descriptive statistics, chi-square tests were used to examine significant differences in personal attributes regarding the hierarchical medical system and the policy to reduce outpatient visits to large-scale hospitals. Regression analysis with two or more independent variables and one dependent variable was applied to predict awareness of the hierarchical medical system and the policy to reduce outpatient visits to large-scale hospitals as well as healthcare-seeking behavior to build a prediction model.

## 3. Results

### 3.1. Analyses of the Basic Information of the Research Subjects

Among the 1340 subjects, there were 779 female participants (58.1%) and 561 male participants (41.9%). Of the female subjects, 15.3, 26.3, and 16.5% were not aware of, partly aware of, or completely aware of the hierarchical medical system and the policy to reduce outpatient visits to large hospitals, respectively; with regard to the male subjects, the percentages of awareness were 11.2, 20, and 10.7%, respectively. In the age group ≥60, most (7.9%) were not aware of the policy, while those who were aware made up the lowest percentage (3.1%); in the age group ≤29, the majority (8.4%) were aware of the policy, while 1.3% was not aware of it. In terms of marital status, most married people were not aware of the policy (15.9%), and those who were single were more aware of the policy (13.1%). Regarding annual income, most people with an income between NTD 200,001 and 400,000 were not aware of the policy (7.3%), while the highest level of awareness was also found in this group (6.9%). By educational background, those with college degrees who were not aware of the policy made up 15.3%; this group also had more subjects that were aware of the policy (16.8%). Because of the sample size, lower percentages of awareness were identified among those with educational backgrounds of junior high, senior high, or graduate school. In terms of occupation, most of the subjects worked for private organizations, and therefore, higher percentages were found for low and high awareness. Participants also included students, military and public servants, and housewives. Table 1 shows the results of personal attributes according to the chi-square test: Gender (*p* = 0.486), age (*p* < 0.001), marital status (*p* < 0.001), educational background (*p* < 0.001), annual income (*p* < 0.001), and occupation (*p* < 0.001).

### 3.2. Analyses of the Impact of the Hierarchical Medical System and the Policy to Reduce Outpatient Visits to Large Hospitals on Residents’ Healthcare-Seeking Behavior

This study further analyzes and compares the impact of the hierarchical medical system and the policy to reduce outpatient visits to large hospitals on healthcare-seeking behavior. Before conducting a regression analysis of the control variables and independent variables, we checked for collinearity by calculating the relevant indices, the variance inflation factors (VIF < 10) and condition index (CI < 10), whose values indicated that there was no problem of collinearity. For the regression model of healthcare-seeking behavior, the *F*-statistic had the value 15.317 (*p* < 0.001). As shown in Table 2 regarding the regression model, for factors impacting healthcare-seeking behavior, the age group of 40–49 (*p* < 0.1), educational level of junior high school (*p* < 0.05), unawareness of the policy (*p* < 0.001), and awareness of the hierarchical medical system and the policy to reduce outpatient visits to large hospitals (*p* < 0.001) reached significance.

Table 2 presents multiple regression estimates for healthcare-seeking behavior. Awareness of the hierarchical medical system and the policy to reduce outpatient visits to large hospitals was a significant predictor, while among sociodemographic variables, age (40–49) and educational level of junior high school were associated with lower levels of healthcare-seeking behavior. These variables explained 17% of the variation in healthcare-seeking behavior.

## 4. Discussion

This study is the first to discuss the relevant influential factors of the hierarchical medical system and the policy to reduce outpatient visits to large hospitals on healthcare-seeking behavior. The results show that about 26.5% of the citizens of Taiwan regardless of gender, age, or educational background were not aware of the government’s efforts to promote the hierarchical medical system and the policy to reduce outpatient visits to large hospitals. The power of policy enforcement determines the public awareness level, as shown in previous studies (e.g., Wu [7]). Wu [7] discussed policy marketing as critical to policy implementation and urged that all political actors should build a consensus through policy marketing for effectiveness.

Second, significant effects were found on healthcare-seeking behavior among different age groups and various educational backgrounds, similar to the findings of Chang et al. [30] and of Shaikh and Hatcher [26]. The age groups of 40–49 years old versus ≥60 years old and those with an educational level of junior high school versus college demonstrate negative significance. It indicates that age has a lower impact on healthcare-seeking behavior in those aged 40–49 years than those aged >60 years, and that the educational level has a lower impact on healthcare-seeking behavior in those who attended junior high school than those who attended university. It is possible that the elderly are more likely to develop chronic diseases and have their own preferences when seeking healthcare. Those with higher educational backgrounds are more likely to use the Internet to search for medical information, which is likely to influence their healthcare-seeking behavior.

Furthermore, low awareness of the hierarchical medical system and the policy to reduce outpatient visits to large hospitals have a significant impact on healthcare-seeking behavior. Similar to the findings of Chang [31], from the perspective of healthcare providers, influential factors of the hierarchical medical system include the negotiation role of medical institutions (44%), policy (22%), connectivity of the medical information system (20%), and healthcare insurance (14%). From the perspective of the public, the hierarchical medical system needs to mainly consider the insufficient capability of physicians (29%), the freedom to choose healthcare (25%), and insufficient clinical equipment (17%). Chen and Lin [32] as well as Hsieh et al. [33] argued that when there is a health problem, the public demonstrates a higher return visit rate to hospitals whose services impressed them. Hospitals have taken more outpatient visits provided by local clinics, harming the medical referral system; hospitals and clinics do not trust each other, which has led to dislocations of Taiwan’s medical resources [12,33]. Lin and Liu [12] also showed that after the introduction of the NHI Act, even patients with stable conditions would choose to return to large hospitals instead of the original local clinics. There is a tendency to pursue higher-level health care services in Taiwan, and the NHI is currently attempting to use national health insurance policy levers to reverse this trend. As a result, if concerns of healthcare provision and the public are not addressed, it will not be possible to successfully promote the hierarchical medical system.

## 5. Suggestions

The empirical results of this study also show that approximately three-fourths of the subjects were not aware or only party aware of the NHI policy. The main purpose of the hierarchical medical system is to lead to changes in healthcare-seeking behavior, improve efficiency, differentiate healthcare services, and promote the division of labor. The decision makers in the NHI must be informed about the re-structuring of the hierarchical medical system and policy to reduce hospital visits when deciding the promotion of policy. Thus, this study suggests that the competent authorities in health administration devote greater efforts to policy promotion to inform the public of the hierarchical medical system to help resolve the issue of healthcare-seeking upgrades.

## 6. Conclusions

Over the past several years, a hierarchical medical system has been the goal of Taiwan’s healthcare policy. Since 2005, the National Health Insurance Administration began promoting a bi-directional medical transfer system in an attempt to increase the return visit rate of patients to local clinics and reduce outpatient visits of patients with minor ailments to large hospitals. In fact, the return visit rate to local clinics remains low. Based on their past visit experience and impressions of healthcare institutions, and especially when the patient–physician relationship is strong, the loyalty of patients toward their physicians will be reinforced [34]. Reasons for a patient to choose a particular hospital include equipment, medical skill, recommendations of friends and families, distance from home, and transportation. With the convenient traffic network today, the distance of travel to regional hospitals and medical centers is often less than ten kilometers [35]. Compared to clinics, large hospitals have more comprehensive medical equipment and diverse medical departments, but this also indicates the weaknesses of Taiwan’s local clinics [30].

The healthcare system and background of each country have different impacts on social development. In Taiwan, although the medical transfer system and family doctors are available, medical resources are not integrated and there have been no coordinative measures. Family doctors fail to play the role of gatekeeper, the public often chooses outpatient visits to hospitals, and it is difficult to avoid treatment of minor ailments at large hospitals. Thus, it is especially important to restrict freedom of healthcare seeking to ensure the implementation of the hierarchical medical system.

## Figures and Tables

**Table 1 ijerph-16-03167-t001:** Relation of the hierarchical medical system and reduce outpatient visits to basic participant variables (*N* = 1340).

Measure	Not Aware	%	Partially Aware	%	Completely Aware	%	*χ* ^2^
Gender							0.486
Women	205	15.3	353	26.3	221	16.5	
Men	150	11.2	268	20.0	143	10.7	
Age (years)							<0.001
≤29	18	1.3	95	7.1	112	8.4	
30–39	57	4.3	131	9.8	109	8.1	
40–49	76	5.7	150	11.2	50	3.7	
50–59	98	7.3	107	8.0	51	3.8	
≥60	106	7.9	138	10.3	42	3.1	
Marital status							<0.001
Single	121	9.0	221	16.5	176	13.1	
Married	213	15.9	368	27.5	174	13.0	
Divorced	21	1.6	32	2.4	14	1.0	
Annual income							<0.001
No income	80	6.0	142	10.6	71	5.3	
≤NTD 200,000	24	1.8	87	6.5	69	5.1	
NTD 200,001–400,000	98	7.3	164	12.2	93	6.9	
NTD 400,001–600,000	73	5.4	130	9.7	73	5.4	
≥NTD 600,000	80	6.0	98	7.3	58	4.3	
Educational level							<0.001
Junior high school	18	1.3	14	1.0	21	1.6	
Senior high school	99	7.4	210	15.7	77	5.7	
University	205	15.3	361	26.9	225	16.8	
Graduate School	33	2.5	36	2.7	41	3.1	
Occupation							<0.001
Students	50	3.7	113	8.4	89	6.6	
Military and public servants	59	4.4	57	4.3	33	2.5	
Private organizations	171	12.8	276	20.6	183	13.7	
Worked	44	3.3	83	6.2	27	2.0	
Others	31	2.3	92	6.9	32	2.4	

**Table 2 ijerph-16-03167-t002:** Multiple regression estimates for healthcare-seeking behavior.

Measure	Healthcare-Seeking Behavior	
	Beta	*t*
Gender (RG: Women)		
Men	0.001	0.043
Age (RG: ≥60)		
≤29	−0.035	−0.744
30–39	−0.001	−0.038
40–49	−0.060	−1.832 *
50–59	0.005	0.161
Marital status (RG: Single)		
Married	0.015	0.418
Divorced	0.008	0.269
Annual income (RG: No income)		
≤NTD 200,000	−0.041	−1.303
NTD 200,001–400,000	0.020	0.397
NTD 400,001–600,000	−0.026	−0.534
≥NTD 600,000	−0.017	−0.345
Educational level (RG: University)		
Junior high school	−0.082	−3.001 **
Senior high school	−0.014	−0.456
Graduate School	0.042	1.543
Occupation (RG: Private organizations)		
Students	−0.064	−1.289
Military and public servants	−0.022	−0.798
Worked	0.050	1.322
Others	−0.019	−0.685
Policy understanding (RG: Partially aware)		
Completely aware	−0.032	−1.166
Not aware	0.119	4.243 ***
Independent variable		
Knowledge of the hierarchical medical system and the policy to reduce outpatient visits to large hospitals	0.405	15.894 ***
*R* ^2^		0.188
Adj. *R*^2^		0.176
*F*-value		15.317
*p*-value		<0.001

Note: *** *p* < 0.01, ** *p* < 0.05, * *p* < 0.1.

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
