# Peer review of "The Impacts of the Hierarchical Medical System on National Health Insurance on the Resident’s Health Seeking Behavior in Taiwan: A Case Study on the Policy to Reduce Hospital Visits"

_ijerph, 2019, doi:10.3390/ijerph16173167_

Round 1

Reviewer 1 Report

This study is interesting for it explores the hierarchical medical system, with a mention to the referral system. It is easy to notice the English writing is smooth and easy to understand. Certainly this paper is organized logically, and data analysis part is clear and easy to catch. Comments are as follows.

Sentence as “Most previous studies focused on the referral system”, etc. are illustrated by author directly, with out a reference, which may not be convincing. Of course author introduces lots of background information about why he chooses this topic. However, this paper didn’t spend pages on the literature review. I’m confused about current studies inner or out of the country. And sentences in line 67-98 is not convincing enough for delivering the literature review.

Author Response

August 19, 2019

Dear Reviewers:

On behalf of all the authors, I am enclosing a revised version of manuscript # ijerph-560418 entitled “The Impacts of the Hierarchical Medical System on National Health Insurance on the Resident’s Health Seeking Behavior in Taiwan: A Case Study on the Policy to Reduce Hospital Visits,” which reflects all your earlier comments.

Herein, we explain how we revised the paper based on those comments and recommendations. We want to extend our appreciation for taking the time and effort necessary to provide such insightful guidance. We have used a professional English editing service to produce this new version of the paper. (Please refer to the certificate about English editing.)

Sincerely yours,

Yu-Hua Yan, PhD

Associate Professor,

Department of Medical Research, Tainan Municipal Hospital, Taiwan ((Managed by Show Chwan Medical Care Corporation).

Department of Hospital and Health Care Administration, Chia Nan University of Pharmacy and Science.

Reviewer 2 Report

Thank you for giving me the opportunity to review this manuscript about the impact of the hierarchical medical system on national health insurance on the resident’s health seeking behavior in Taiwan. The paper addresses the research through a survey conducted among 1,340 health-seekers and analyses the findings with a regression model. The research identifies the factors that are significant and concludes that policymakers should invest more in healthcare promotion to increase people’s understanding of the hierarchical medical system.

The paper explores an interesting topic, which is quite new also considering the setting to which it is applied, as the process under investigation has started in 2018. While I see the potential of the paper, I have several issues with this manuscript which I think should be taken into consideration by the authors.

Introduction:

1) I have concerns related to the literature adopted, as about the 50% of the literature considered in the references is in Chinese. Although I understand the link of the literature adopted with the case under study and the specific setting, I believe that this could threaten the theoretical relevance of the paper to an international audience. I would suggest to the authors to deepen the literature search and to include relevant works written in English. This may also help the authors in different ways:

-        To support some statements that at the moment appear to be lacking in evidence (e.g. p. 1 line 35 “Unnecessary healthcare-seeking behavior also leads to a waste of medical resources”)

-        To better frame the contribution of the paper, on the light of the gaps detected in the literature (please, see following comments).

2) Concerning the context to which the study is applied and the reforms is undergoing, the paper would benefit from a discussion at the beginning of p. 2 the characteristics of the medical referral system and hierarchical medical system (and the bi-directional referral system mentioned at p.2 line 74). This would help the readers in fully understand the setting.

3) While reading the introduction, it appears to me that the aims of the study are not clear. The authors state in the abstract “This study investigated the effect of the hierarchical medical system under the national health insurance program on resident’s healthcare-seeking behavior in Taiwan” and in the introduction “This study investigated the effect of the hierarchical medical system under the NHI program on healthcare-seeking behavior in Taiwan”. What do the authors mean by “effects”? What are the research questions that drive the research? The introduction is quite descriptive and not fully theoretically informed. I would suggest to enrich the literature search to drive a better definition of the research questions; this would allow the reader to better understand the rest of the paper.

4) The authors should better explain the gaps in the literature, and the final contribution of the paper. In the introduction, the authors state: “The research investigating the novel policy ‘allowance reduction of outpatient visits’ in Taiwan is still lacking. Most previous studies focused on the referral system rather than the effects of the hierarchical medical system on the healthcare seeking behavior of Taiwanese residents”. Could the authors provide support for those claims? The enrichment of the literature search could be helpful also these regards.

5) There are sentences that are not adequately supported by references, in particular those referring to the political situation in Taiwan: e.g. “With regard to the current NHI policy in Taiwan, however, government officials only care for votes, and no one is willing to engage in reform” (p. 2 line 70-71) (this in particular it seems a quite strong sentence if not adequately supported); “The NHI program in Taiwan guarantees the right of people to seek healthcare regardless of their financial status and the freedom to choose healthcare providers. Because of fierce competition and fees for services in the system, medical institutions aim to increase the number of cases attended to” (p.2 line 88-90); “Healthcare authorities in Taiwan initiated the allowance reduction of outpatient visits at regional and higher hierarchical hospitals in 2018. As per this, the allowance of outpatient visits at regional hospitals and medical centers will be reduced by 2%, followed by a total of 10% reduction over 5 years. The ultimate goal is to implement a hierarchical medical system and provide the residents accessible as well as consistent medical services” (p. 3 line 99-103). Are there researchers, government reports, etc. that support those claims?

Methodology:

1) The methodological section should be improved. For instance, the research sample is not adequately presented. In section 2.1 the research sample and the source of data are presented. The research sample is composed by “subjects with recent healthcare-seeking records”: How were the names of the subjects involved in the study collected? From which database/source of information? Authors should also deepen how the purposive sampling technique was used.

2) In addition, in the introduction section, the authors report that 1,000 questionnaires were released in September and October 2018, but in the abstract and in the method is reported that the questionnaires were collected between August and December 2018, for a total of 1,340 questionnaires: How many questionnaires were submitted? When? How many people were contacted in total? How many accepted to fill the questionnaire? Which percentage of valid questionnaires resulted over the total amount of questionnaires submitted?

3) The research tool adopted is based on Berkowitz and Flexner (1981) research, which deepened the perspectives of the consumers with a personal physician in the US setting. The authors, however, do not discuss the variables and the items included in the questionnaires, as the paper is briefly mentioned only in the introduction section. An in-depth discussion of the variables included in the questionnaire (and the items considered) in the method section is needed in order to allow the reader to understand the model of the study, and then to appreciate the results, the discussion and the contribution. In this sense, providing also the questionnaire would be helpful.

Results:

Results are presented in a convincing way, but with the support of a description of the questionnaire and a discussion of the variables included in the study the reader might gain more insights from the case and better understand the discussion and the conclusions.

Discussion:

Concerning the discussion, I suggest the authors to present in the introduction section the papers that are then included in section 4 and that are used to support the discussion. This would help the authors to depict a more theoretically informed introduction and to provide an overview of the researches in the field that guide the definition of the research questions (see comments 3-4). The discussion section could benefit from this, as the authors could then discuss their results on the light of the relevant literature already presented.

Conclusions:

The conclusions seem to take a while to get to the actual contribution of the work and to the implications. I suggest the authors to better stress how the paper contributes to the literature, based on the gaps that will be further detected revising the literature search in the introduction section, and to present the implications for policy makers at national level and perhaps for managers with the organizations.

Also in the abstract, the conclusions should be rewritten to shortly explain the theoretical contribution of the paper and its practical implications.

Overall, I enjoyed reading the paper and whish the authors the best for the following steps of their research.

Additional comments:

a)      Author contributions: the names of the 3 authors do not fully match the names reported in the author contributions (Yu-Hua Yan, Chih-Ming Kung, Horng-Ming Yeh vs Yu-Hua Yan, Liang-Hsi Kung).

b)      “Safety reserves are expected to b used up by 2120” (p. 1, line 40): did the authors meant 2020?

c)      P. 3 line 133 “A questionnaire pretest using 30 copies was conducted onsite before formal interviews…”. What do the authors mean by “onsite”?

Author Response

(The authors gave the same response as above.)

Round 2

Reviewer 2 Report

Dear authors,

thanks for the efforts in reviewing the paper.

I believe you have done a good job and addressed the comments. The introduction and the aims of the study are now clearer and consistent with the results and discussion section, while the methodology is presented in a detailed way.